# Talk Less, Interact Better: Evaluating In-context Conversational Adaptation in Multimodal LLMs

**Yilun Hua and Yoav Artzi**
Department of Computer Science and Cornell Tech
Cornell University
{yilunhua, yoav}@cs.cornell.edu

## Abstract

Humans spontaneously use increasingly efficient language as interactions progress, by adapting and forming ad-hoc conventions. This phenomenon has been studied extensively using reference games, showing properties of human language that go beyond relaying intents. It remains unexplored whether multimodal large language models (MLLMs) similarly increase communication efficiency during interactions, and what mechanisms they may adopt for this purpose. We introduce ICCA, an automated framework to evaluate such conversational adaptation as an in-context behavior in MLLMs. We evaluate several state-of-the-art MLLMs, and observe that while they may understand the increasingly efficient language of their interlocutor, they do not spontaneously make their own language more efficient over time. This latter ability can only be elicited in some models (e.g., GPT-4) with heavy-handed prompting. This shows that this property of linguistic interaction does not arise from current training regimes, even though it is a common hallmark of human language.

## 1 Introduction

Human interlocutors adapt to each other during interactions, developing increasingly efficient ways to refer to concepts and objects. Hawkins et al. (2020b) exemplify this via communication between a nurse and a bed-ridden patient at home. Initially, the patient may refer to a medicine with *the medicine for my back pain in a small blue medicine bottle ...*, but after a week of care, they are likely to just ask for their *back meds*. This increase in efficiency relies on the interlocutors forming ad-hoc linguistic conventions: the mutually understood, concise phrases to communicate referential content. This phenomenon has been repeatedly observed and characterized in controlled studies using repeated reference games (Figure 1; e.g., Krauss & Weinheimer, 1964; Brennan & Clark, 1996; Hawkins et al., 2020a).

We study this ability in multimodal large language models (MLLMs). LLMs and MLLMs are well positioned to acquire this behavior and display it spontaneously in interactions. They are trained on large amounts of human language data, in which this behavior is common and the history of an ongoing interaction is often retained, thereby explicitly keeping the information needed at hand. Beyond the scientific question, such ad-hoc adaptation has significant application impacts: enabling more natural interactions, reducing the costs involved in conversations (e.g., using shorter utterances to communicate the same amount of information), and increasing the accuracy of relaying intent.

We propose ICCA,[1] an automated framework to evaluate and characterize the ability of models to form ad-hoc conventions. ICCA uses a corpus of human-human reference game interactions, allowing for completely automated evaluation, which does not require further human interaction, making it easy to deploy for the analysis of new models. The interaction follows the standard repeated reference game setup (Clark & Wilkes-Gibbs, 1986), where a speaker refers to an image within a shared context of images, and a listener resolves the

---

[1]ICCA stands for In-context Conversational Adaptation.

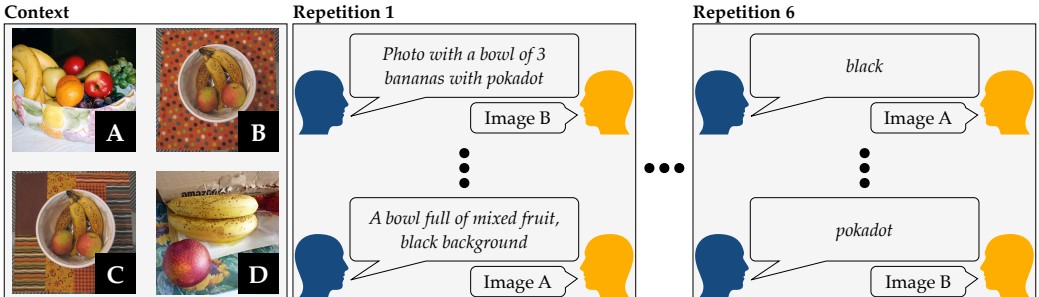

Figure 1: Illustration of a reference game. The speaker (blue) and listener (orange) observe a shared set of images.[2] The interaction progresses in six repetitions, each includes a trial for every context image. In each trial, the speaker describes a target image, and the listener has to select the correct target given the description only. For simplicity, this figure omits the feedback on listener actions. This interaction illustrates some of the effects of convention formation: the descriptions become shorter as the interaction progresses, and lexical choices converge to a subset of the words used in earlier repetitions.

reference to select one of the images, ideally the one originally referred to. Figure 1 illustrates the scenario. We focus on in-context adaptation – as the interaction progresses, the entire history is retained in-context. Core to our approach is comparing the changes in model behavior, either as speaker or listener, throughout an interaction to the changes observed in humans. We measure different properties that have been shown to be influenced by convention formation: utterance length, lexical convergence, and selection accuracy.

We apply our approach to five representative MLLMs: IDEFICS (Huggingface, 2023), LLaVa-1.5 (Liu et al., 2023a), GPT4-vision (OpenAI et al., 2024), Gemini 1.0 Pro Vision (Google, 2023), and Claude 3 opus (Anthropic, 2024). We find that all models struggle to spontaneously introduce conventions and adapt as speakers. Prompt engineering an explicit in-context instruction specific to the reference game scenario can address this to some degree. The strongest models (GPT4, Gemini, and Claude) can then gradually use shorter messages (gaining lexical efficiency) but still struggle with convergence or stability, which hinders the emergence of truly efficient communication. When acting as a listener, GPT4 displays adaptation trends close to humans, improving its accuracy as the interaction progresses, while other models show this behavior to a lesser degree or only under some simplified setups. Overall, we show that while today's MLLMs may passively understand the evolving language of their interlocutor, the ability to adapt their own language for efficient communication does not naturally emerge from their training or instruction-tuning. This outlines important future research problems. We release ICCA under the MIT license at `https://github.com/lil-lab/ICCA`.

## 2   Background and Related Work

**Repeated Reference Games**   A reference game is an interaction where a speaker and a listener (i.e., a dyad) interact over a shared context. The shared context is a set of images. The speaker describes a target image. The target designation is only revealed to the speaker. The listener has to select an image following the speaker's description. Each participant sees the images in a different order, so they cannot use position information to communicate the referent. Reference games have been used extensively in the study of computational models, including recently to evaluate visual abstraction (Ji et al., 2022) and conversational aptitude (Chalamalasetti et al., 2023).

A repeated reference game (Figure 1) includes multiple repetitions. Each repetition has one trial for each image in the shared context. The listener receives feedback after every

---

[2] In Figure 1, we show the shared context only once for compactness. In our experiments, we shuffle and show the context for each trial, but also experiment with showing it only once.

trial, which indicates the correct selection. The repetition and feedback allow the dyad to form message agreements over the repeating stimuli (i.e., the speaker would naturally use gradually shorter but related messages across repetitions, and the listener learns what they refer to). ICCA's repeated reference games are developed based on the setup and data from Hawkins et al. (2020b). The shared context includes four images, and there are six repetitions, giving a total of 24 trials. The order of images is shuffled across the trials. We refer to this as the *standard setup*. Hawkins et al. (2020b) only uses this *standard setup*. Beyond this standard setup, we design variants to further disentangle the types and causes of model failures for in-context LLM adaptation (Section 4 and Section 5).

**Ad-hoc Adaptation in Interactions**    Existing literature shows that humans are inclined to reduce the effort needed to convey their intended information and for their audience to comprehend it, leading to efficient communication (e.g., Zipf, 1949; Gibson et al., 2019; Yin et al., 2024). When human individuals interact through dialogue, this is manifested by developing and using ad-hoc linguistic conventions. This phenomenon has been observed with repeated reference games (Krauss & Weinheimer, 1964; 1966; Clark & Wilkes-Gibbs, 1986; Hawkins et al., 2020a), and related interaction scenarios (Haber et al., 2019). Studies have also shown various properties of these conventions, such as arbitrariness, stability, stickiness, and convergence (Lewis, 1969; Brennan & Clark, 1996; Markman & Makin, 1998; Hawkins et al., 2020a; Eliav et al., 2023). This adaptation was modeled with the pragmatic rational speech act model (RSA; Goodman & Frank, 2016), leading to development of models that replicate this behavior in reference games (e.g., Monroe et al., 2017; McDowell & Goodman, 2019; White et al., 2020) and use it to improve model performance on other tasks (e.g., Andreas & Klein, 2016; Fried et al., 2018). Adaptation was studied beyond reference games, showing how the complexity of the scenario influences how conventions manifest in the language (Effenberger et al., 2021).

Ad-hoc conventions are a particular instantiation of the broader phenomenon of common ground, which is defined as the mutually recognized shared information between the participants of a conversation (Clark & Brennan, 1991; Lewis, 1969; Stalnaker, 2002). Common ground has been studied extensively, with focus on both human cognition (Clark, 1996; Horton & Gerrig, 2016) and machine reasoning (Cohen & Levesque; Grosz & Sidner; Traum, 1994; Del Tredici et al., 2022; Shaikh et al., 2024; Andukuri et al., 2024; Testoni & Fernández, 2024).

**Model Adaptation**    Adapting models during an interaction to improve communication efficiency or success is relatively understudied. Hawkins et al. (2020b) proposes a continual learning method for CNN-RNN models to gain communication efficiency in repeated reference games through continual weight updates. Zhu et al. (2021) proposes explicitly training models for ad-hoc adaptation through meta-learning. We focus on the in-context capabilities of LLMs and MLLMs, which offer an update-free route for adaptation that is particularly compelling given the costs of updating large models. Our use of in-context learning differs from how this mechanism is usually used either by providing instructions (Ouyang et al., 2022) or few-shot examples (Brown et al., 2020). While reference games can be seen as related to the few-shot approach, ad-hoc adaptation is not about replicating patterns, but showing change and adaptation over time.

## 3   The ICCA Framework

ICCA uses a dataset of human-human interactions, and allows to easily customize different parts of the interaction. This flexibility enables different research questions. For example, in Section 5, we customize the interaction structure to analyze how well models handle long interactions with multiple images interleaved in them. ICCA supports studies with the model acting either as speaker or listener, and includes several metrics to track different properties of adaptation during the interaction.

ICCA is fully automated and easily applicable to new MLLMs. Our design does not require collecting new data or human studies, but instead uses Hawkins et al. (2020b)'s human-human interaction data to simulate a human interacting with an MLLM. Each

interaction in the dataset was collected under the standard setup (Section 2) and uses a visually challenging reference context, consisting of four similar images (Figure 1). The dataset contains 54 human-human interactions, which we incorporate into ICCA.

A repeated reference game interaction $R$ is a sequence of tuples $\langle (C_i, c_i, s_i, l_i, f_i) \rangle_{i=1}^n$, where $C_i$ is the set of images forming the reference context, $c_i$ is the index of the target image, $s_i$ is the speaker utterance, $l_i$ is the listener selection, and $f_i$ is the feedback based on the listener's selection. At every trial $t$, with the evaluated model as either the listener or speaker, ICCA constructs the MLLM prompt from an instruction text $I$, the history (i.e., all prior trials $R[:t]$), and the stimuli for the current trial with a user-defined pre-processing function $F$. Upon receiving the model's response, ICCA computes the feedback $f_t$ to help the next trial.

The function $F$ prepares the prompt depending on the experiment, and is key to the flexibility of ICCA. For example, when evaluating a model as a listener under the standard setup, the function input would be $F(I, R[:t], C_t, s_t)$, and it would format and concatenate all the elements in order. Figure 7 in the appendix shows an example prompt. ICCA allows to easily modify this standard setup, for example by processing the data such that $F$ drops all reference contexts except $C_1$, thereby creating a simpler input where the images appear only once at the beginning of the interaction.

ICCA simulates the interlocutor of the model evaluated either with a deterministic counterpart or by having another model take the role. A deterministic speaker outputs the messages from recorded human interactions dataset, showing predetermined, realistic trajectories of message shortening over time, but it does not adapt its language based on the listener's selections. It can be considered as a "convention comprehension" task, potentially more challenging due to the non-adapting speaker messages. We use this simulated speaker for our model-as-listener experiments (Section 5) because it exposes the model listener to behaviors and linguistic conventions naturally occurring in human interactions. Inversely, for our speaker experiments (Section 4), we use a high-performance model listener (GPT4), which we observe to have performance similar to human listeners (Section 5).

We evaluate model behavior with adaptations of the metrics used in human studies with reference games (Hawkins et al., 2020a;b). For listener experiments, we follow Hawkins et al. (2020b) and report the average accuracy in each repetition. Speaker experiments are evaluated using several metrics. We report the average message length and the listener's accuracy in each repetition. Additionally, we evaluate the similarity between corresponding messages from consecutive repetitions. While this was done using GloVe embeddings (Pennington et al., 2014) in past work (Hawkins et al., 2020a), we design a new metric called Word Novelty Rate (WNR), which is sensitive to exact word choices. WNR is a modified word error rate that only counts insertions and substitutions, and ignores deletions. It is motivated by how people naturally drop words from their messages as the interaction progresses (Hawkins et al., 2020a), whereas additions and substitutions of words often reflect important changes in information based on our observations. Compared to GloVe, WNR is more sensitive to lexical inconsistencies that can increase the listener's cognitive load. Appendix A presents more details of our metrics, including a comparison of WNR and embedding-based similarity metrics and a variant of WNR that is not normalized by message length (referred to as Word Novelty Distance).

## 4  Model-as-speaker Experiments

We study model behavior as speaker with five state-of-the-art vision MLLMs: IDEFICS-80b-instruct,[3] LLaVa-1.5-13b, GPT4-vision, Gemini 1.0 Pro Vision, and Claude 3 opus. Throughout all speaker experiments, we customize the data to only show the referential context once at the beginning of the interaction, so there is no shuffling of context throughout the interaction. We engineer prompts for each model individually to best evaluate its capability. We use GPT4 as the listener. It exhibits high performance in our listener experiments (Section 5), especially when the context appears only once at the beginning,

---

[3]IDEFICS is an open-source reproduction of Flamingo (Alayrac et al., 2022).

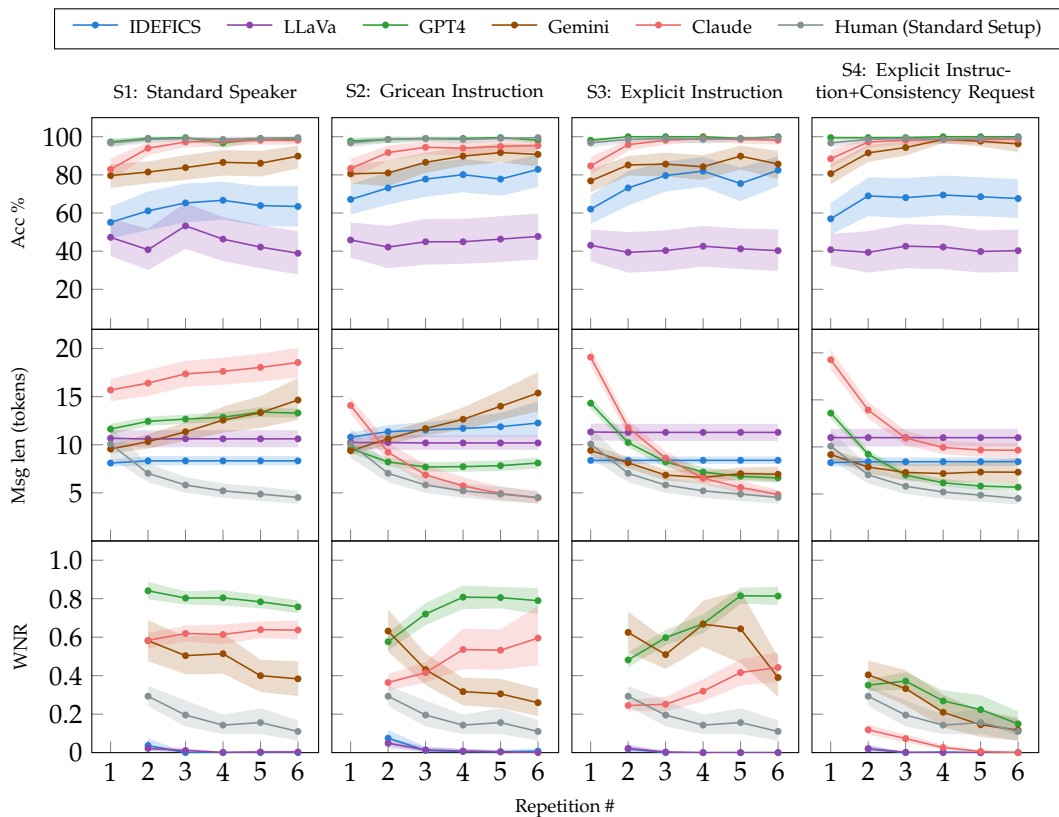

Figure 2: Speaker experiments. Margins of errors are bootstrapped 95% CIs.

so it is a suitable substitute for a human listener in this study. Appendix A.3 provides implementation details.

We design four speaker variants by modifying the instruction $I$. The variants were developed throughout our experiments, by observing the difficulty of models to present patterns similar to human linguistic behavior. The variants instruct the model to display the convention formation behavior observed in human speakers in increasingly explicit and specific ways:

**S1: Standard Speaker** The standard speaker setup (Section 2). The model speaker only receives the basic game instruction, with no mention of communication efficiency. Figure 6 in the appendix shows an example prompt.

**S2: Gricean Instruction** A relatively light-handed and general way to introduce the expected convention formation behavior is to explicitly instruct the model to follow the Gricean quantity maxim. This kind of instruction is not specific to reference games, and does not explicitly mention message length. Its focus is information, and it entails that cooperative interlocutors would provide enough information to identify the referent but would not make the message more informative than necessary. We add additional instructions based on the maxim and further instruct the model to *think about how the amount of information needed may change as more trials are completed and based on the listener's performance in previous trials*.[4,5]

**S3: Explicit Instruction** We instruct the model to explicitly reduce message length as the interaction progresses. Unlike S1 and S2, this instruction is specific to reference games, as language adaptation in other scenarios is not necessarily accompanied by length reduction (Effenberger et al., 2021). We add to S1 an explicit instruction to

---

[4]The models did not show substantial improvement without this additional instruction.
[5]Not the exact prompt used for experiments; shortened and revised for illustrative purposes.

reduce utterance length: *as more trials are completed and as the listener understands you better, gradually condense your messages, making them shorter and shorter every trial*.[5]

**S4: Explicit Instruction + Consistency Request** Convention formation in reference games is not only characterized by reduction in utterance length, but also by lexical consistency. This variant explicitly instructs the model to follow this pattern. Similar to S3, it is specific to the repeated reference game setup and its use of a repeating context. We add to S3 the instruction: *when creating a shorter message for an image, try to extract salient tokens from the previous messages for this image rather than introducing new words. The short messages should still allow the listener to choose the target correctly. For each image, when you reach a message that cannot be further shortened, you should keep using that message for the rest of the game*.[5]

Figure 2 shows the results for all variants, along with properties of the human messages from Hawkins et al. (2020b), which were collected using the standard setup. We report mean message length, WNR, and listener accuracy for each repetition. Overall, all models fail to spontaneously improve communication efficiency. It is only with fairly heavy-handed instruction that GPT4, Gemini, and Claude show adaptation trends similar to humans.

Variant S1 shows that without any explicit instruction, the models show trends that are far from human behavior. GPT4, Gemini, and Claude generate longer utterances in later repetitions. IDEFICS and LLaVa maintain consistent message lengths. But, upon close inspection, we observe they simply tend to repeat previously used messages for the same image. This explains their very low and almost constant WNR. We analyze this further in Section 6. Also, the messages of IDEFICS and LLaVa are less effective in distinguishing the target images, as shown by the lower listener accuracies. This demonstrates the inability of these models to correct their behavior based on feedback. No models show WNR trends similar to humans. GPT4, Gemini, and Claude constantly introduce new words as the game progresses, as shown by higher WNR curves, even though we do not use token sampling for decoding. Gemini shows a downward trend, but achieves this by the undesirable practice of making its messages longer every trial while making a relatively constant number of word insertions or changes, so a bigger portion of the message is maintained.[6]

Gricean instruction (S2) leads GPT4 and Claude to reduce message length over time, though GPT4's reduction is far from humans'. Both models have WNR curves significantly higher than humans. As their messages shorten, they still frequently introduce new words and do not stabilize the messages, a behavior adverse to communication efficiency. We further discuss this issue with S3, where more models display this issue.

S3's explicit instruction has no impact on IDEFICS and LLaVa. Both continue repeating messages, failing to follow the instructions. Gemini and GPT4 show decreasing message length trends similar to humans but still have longer messages than humans throughout. Claude eventually produces messages as short as humans but starts with much longer messages than other models. All models showing message shortening frequently introduce new words as they shorten the messages. Even when the message is too short to be condensed, the models may adopt new words next time without changing the message length. Figure 8 in the appendix exemplifies these behaviors. Such behaviors deviate from the stability property of conventions and the observation that human messages show high consistency and convergence. The gap between the WNR curves of these models and humans illustrates this issue. While these models show increasing lexical efficiency, the use of new words reduces communication efficiency, burdening the listener to reason about the words that did not appear the last time the image was referenced. We further discuss this issue in Section 6.

S4 addresses the consistency issue but requires further explicit instruction. The final prompt elicits from GPT4, Gemini, and Claude both length reduction and message convergence, as observed with humans. However, the S4 prompt is very specific to ICCA's setup and does not generalize beyond reference games. Heavy-handed prompt interventions like this are

---

[6]This is due WNR's length-normalization, which is critical to reflect similarity (Appendix A.2).

also known to cause unintended model behaviors (Shaikh et al., 2024). Prompt engineering is not likely to be the solution.

# 5 Model-as-listener Experiments

Listener experiments follow a setup similar to the speaker experiments as far as models and prompt optimization (Section 4). Gemini, LLaVa, and Claude cap the number of input images, limiting their use in some of our listener variants. Overall, we design four main variants, each implemented through the pre-processing function $F$. Our design process is iterative, with some variants designed based on the behavior observed with earlier ones. Throughout the listener variants, we keep the instruction $I$ largely constant and about the role of the listener. We vary how we display the referential context.

The listener action space is more limited, simply requiring the model to select the referenced image. We focus on evaluating model accuracy, similar to how listener behavior is characterized in human studies. Figure 3 visualizes the behaviors we observe. We also include human listener accuracy trends as reference to model accuracies.

The starting point for the listener study is the standard reference game setup (Section 2):

**L1: Standard Listener** Images are shuffled and re-displayed for each trial, so each image will potentially have a new label relative to previous trials.

L1 requires a growing number of images in the prompt as the interaction progresses. With six repetitions of four trials and a context of four images, the maximum number of images in the prompt at the end of the interaction is 96. Gemini, LLaVa, and Claude can take at most 16, 4, and 20 images, so we only use GPT4 and IDEFICS with L1.

We expect an effective model to exploit the conversation history to reason about the human speaker's conventionalized ways of speaking. Even if the model starts with low accuracy, it has the opportunity to improve because the prompt at later stages includes feedback for its choices, and as the messages conventionalize, later messages for an image are often exact repetitions, albeit with the referential context shuffled.

Both GPT4 and IDEFICS do significantly worse than humans (Figure 3, left). As expected, humans demonstrate strong performance to start with, and show an upward trend in accuracy, as the interlocutors adapt to each other. GPT4 is significantly worse than humans, though performing fairly well. It shows a marginal improvement trend (88.9%→92.5% in repetition 5), but it is not significant, and weakens in the last repetition (91.2%). IDEFICS is much worse immediately in the beginning (46.8%), and rather than improving, its performance deteriorates as the interaction progresses, reaching random chance in the later trials. This happens even though it is receiving an increasing amount of information that should allow it to improve its performance. A possible cause for this trend is the dramatic increase in the prompt size, especially as more and more images are added, as the interaction progresses. We further discuss this issue and its potential causes in Section 6.

## 5.1 History and Context Impact

Following the observations with L1, we design three variants with simplified referential contexts and history to better understand how well the models handle the interaction history and the referential context:

**L2: No History** Each of the 24 trials is given to the model in isolation, without any history. The model input includes the context of four images and the speaker utterance, as well as the basic game instruction. This variant reveals the extent to which the model can reason about the ad-hoc conventions formed in repeated human-human interaction without access to the history in which they were formed.

**L3: Images Once** A potential challenge of L1 is the large number of images in the prompt. A model's architecture or training may not be suitable for handling a large number of images, and LLM prompt length is known to adversely influence

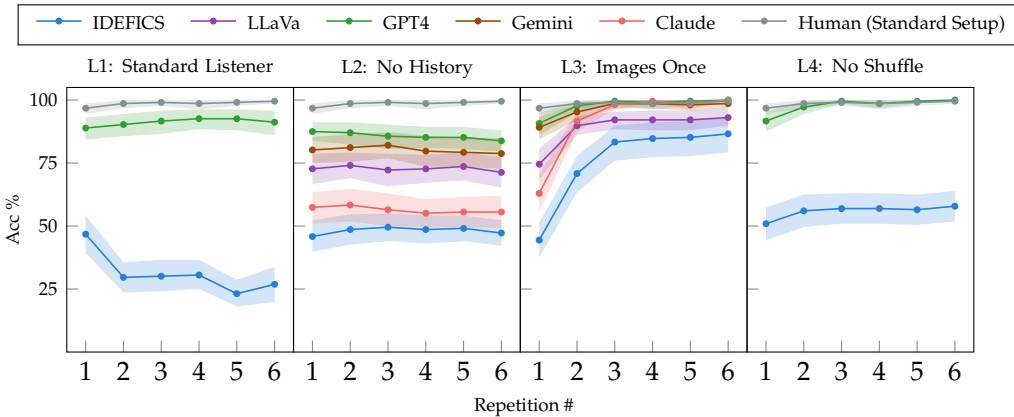

Figure 3: Listener experiments. Margins of Error are 95% bootstrapped CIs.

performance (Liu et al., 2023b). L3 uses a shorter history, by only providing the referential context (four images) to the model once in the first trial. Image labels are persistent across all trials, which also avoids the impact of shuffling. Unlike L2, this variant includes the complete message, selection, and feedback history.

**L4**: **No Shuffle**   The four images appear every trial similar to L1 but are not shuffled across trials. L4 shows the effects of image shuffle if compared with L1. It also shows the effect of image quantity if compared with L3, which does not involve shuffling either.

L2 and L3 need only four images in the prompt, allowing us to test all the models.

The no-history variant (L2) reveals different performance trends for IDEFICS and GPT4 compared to the standard setup (L1). IDEFICS is still not doing great (45.8% on the first repetition), but performance largely remains consistent across repetitions, indicating that possibly the complexity of the prompt is at the root of its downward trend in the L1 scenario. GPT4 starts with similar performance (87.5%) to its results in L1, but then shows a slight downward trend (83.8% at the end), in contrast to the initial upward trend in L1. This indicates that the gradually conventionalized, shorter messages a human speaker uses tend to be more difficult for the model to resolve and that the conversation history can be an effective remedy. Gemini (80.2→78.8%), LLaVa (72.7→71.3%), and Claude (57.4→55.5%) show similar trends to GPT4 from Repetition 1 to 6, with overall lower accuracies.

The benefit of history becomes more conspicuous when the referential context is only given once (L3). This variant dramatically simplifies the prompt, but retains the information needed for convention formation (i.e., prior message, selection, and feedback). All the models show an upward trajectory as the interactions progress. GPT4 and Claude are the strongest, eventually reaching 100% accuracy, matching human listeners' performance (99.54%). This suggests that all models can associate the current message with the relevant prior messages, thereby increasing their prediction accuracy as the interaction progresses. Admittedly, because an image always has the same label throughout the game under this setup, the models may do well by simply drawing associations between an image's label and the messages that have referred to that image (i.e., *label-message* associations). Under this mechanism, the model can improve its accuracy without reasoning about the actual visual input. We explore this possible mechanism with more game variants in Appendix D. Nonetheless, this experiment shows that MLLMs possess some capabilities for increasingly efficient communication with humans when acting as the listener.

The no-shuffle (L4) experiment also provides key insights. GPT4's accuracy is similar to that in L3 and higher than that in L1. This demonstrates sensitivity to image shuffling and the constantly changing image labels. GPT4 in L3 and L4 may be relying to some degree on text similarity between repetitions by exploiting the label-message associations, rather than grounding to the visual input. We study this further in Appendix D. IDEFICS' performance

on the other hand is different from both L3 and L1, showing an almost unnoticeable trend up. This shows that it is not only the shuffling but also the increase in the number of images that IDEFICS cannot seem to handle well. We further discuss this issue in Section 6.

# 6 Discussion

Our studies point to various issues that likely hinder specific models from displaying communication efficiency gains, and point out directions for future works.

**Tendency to Repeat Messages**    In the speaker study, IDEFICS and LLaVa tend to repeat the first message they use for each image, showing no adaptation. To further study how much these models prefer patterns of repetition, we design a test that uses these models for language modeling rather than text generation. We construct two transcripts for each of the 54 human-human interactions in our original dataset. One is the original transcript from human-human interactions, showing the natural evolution of messages. The other is a manipulated transcript where the speaker repeats the messages from Repetition 1 in all the later repetitions. We calculate the log probability and perplexity IDEFICS and LLaVa assign to these transcripts, count the number of times one type of transcript has better log-probability/perplexity than the other, and apply a sign test. We find that the manipulated transcript showing message repetitions consistently receives higher log probability and lower perplexity for all 54 interactions, showing the models' significant tendency towards repeated patterns (sign test p-values are near zero). Unfortunately, this experiment cannot be done with GPT4, Gemini, or Claude due to API limitations.

**Lexical Efficiency ≠ Communication Efficiency**    The convergence of human speaker messages for a particular image to a short, stable convention often takes the form of extracting salient tokens from the previous message and sticking to the same message once it becomes very short (Hawkins et al., 2020a). Unless directly instructed to do so through a highly engineered prompt (S4), GPT4, Gemini, and Claude often introduce new words when shortening their messages or even when the messages cannot be further shortened, as shown in the S3 explicit instruction variant. Such inconsistency with human behaviors is problematic. When messages for the same image do not converge, no conventions can form and the listener will likely need additional cognitive effort to process the previously unseen words. Intuitively, even if a new message is semantically similar to a previous one by using synonyms, resolving it still likely entail a greater cognitive load than an exact repetition. Moreover, when new words describing a new aspect of an image are introduced after a few rounds of relatively similar messages, they violate the human listener's expectation, potentially leading to miscommunication and slower response (Metzing & Brennan, 2003).

**Performance Degradation with Many-image Inputs**    Among the models that support a large number of images, IDEFICS performs much worse as the number of images increases. Even though the images in L4 are not shuffled across trials, which could have allowed the model to exploit label-message associations as an efficient way to gain high accuracy, the model still had much lower accuracy than when the images only appear once (L3) (Figure 3). When the history contains a growing number of images that are shuffled between trials (L1), IDEFICS shows an even worse accuracy trend.

A likely hypothesis is that a greater number of images creates challenges for capturing the dependency between specific visual input and textual cues, which can manifest as failures to associate an image's label with the actual content of the image. In a qualitative experiment, we supply a sequence of images and their labels as input to IDEFICS, and instruct it to describe *Image [X]*. We observe that IDEFICS can describe the correct image easily when we give up to four labeled images, but often makes mistakes as the number increases (Figure 9 in the appendix). Therefore, even though IDEFICS is designed and trained to support multi-image inference,[7] its multi-image capabilities do not generalize beyond a few images.

---

[7]The Flamingo architecture behind IDEFICS supports an arbitrary number of images as input and it is trained on interleaved texts and multiple images (Huggingface, 2023).

Another potential cause is the Flamingo architecture behind IDEFICS. Each token's cross-modal attention only applies to the visual features of the last image that precedes it, rather than all the images. Information about other images can only be indirectly accessed through self-attention on the hidden states of their respective <image> tokens in the text sequence. This architecture may degrade the ability of the model to reason about images, depending on their locations in the input. When the target is not the last image, Image D, the message tokens will not have direct cross-modal attention to the target's visual features, which may hurt IDEFICS' prediction. In L3, where IDEFICS did perform well, this limitation might have been mitigated by the strong textual cues and label-message associations, whereas having more images may distract IDEFICS from these cues and thus manifests this limitation.

# 7   Conclusion

ICCA provides a perspective into the performance of today's MLLMs that is missing in existing benchmarking, and can be easily applied to new MLLMs without collecting new human data. We observe that state-of-the-art models lack the in-context abilities to adapt their own language for efficient communication, even though they may sometimes perform better while passively receiving increasingly efficient language from their interlocutor. This issue is fundamental because, unlike humans, the models do not perceive the effort or cost needed for communication, thus having no inherent reason to reduce them. It is still surprising though, given that LLMs/MLLMs have successfully displayed many other human behaviors and impressive abilities in various applications, by learning from the large amounts of human data, where adaptation for efficiency is common. Overall, the current paradigm for creating LLMs fails to address the need for conversational adaptation and future research is needed on improving their abilities to spontaneously improve language efficiency, maintain language consistency for the same referent, avoid excessive tendency for repetitions, and handle more images in a single query.

### Acknowledgments

This research was supported by ARO W911NF21-1-0106, NSF under grant No. 1750499, a gift from Open Philanthropy, and a gift from Apple. We thank Robert Hawkins and Marten van Schijndel for insightful discussions. We thank the anonymous reviewers and the area chair for their valuable feedback.

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

# A Implementation Details

## A.1 Message Length Measurement

We measure the length of the generated messages by counting the number of tokens. Because different MLLMs have different tokenizers, we choose to only use IDEFICS's tokenizer for message length calculations for all models.

## A.2 Word Novelty Rate

Word Novelty Rate is a modified Word Error Rate, which only counts insertions and substitutions, and ignores deletions. The number of insertions and substitutions is normalized by the length of the reference message, as done in the standard Word Error Rate calculation. For two messages from Repetition N-1 and N, we use the message from Repetition N-1 as the reference and the one from Repetition N as the hypothesis. We follow Hawkins et al. (2020a)'s metric design and drop most function words to only consider open-class content words (nouns, adjectives, verbs, and adverbs) as well as pronouns, numbers, and adpositions.

WNR addresses the limitation of metrics using cosine similarity between averaged embeddings (e.g., GloVe), which operate in the semantic space, for example as in Hawkins et al. (2020a). Semantic similarity between messages is not sensitive to some lexical changes, ignoring the importance of exact word choice in convention formation. For example, once a convention is formed to use *the car* to refer to an image, changing it to a semantically similar message, *the automobile*, violates the stability property of conventions, which may increase the listener's cognitive load. Empirically, we find that WNR produces results consistent with Hawkins et al. (2020a)'s averaged GloVe embedding similarity, as shown in Figure 4 (WNR moves in the opposite direction to the GloVe-based similarity because WNR directly measures dissimilarity).

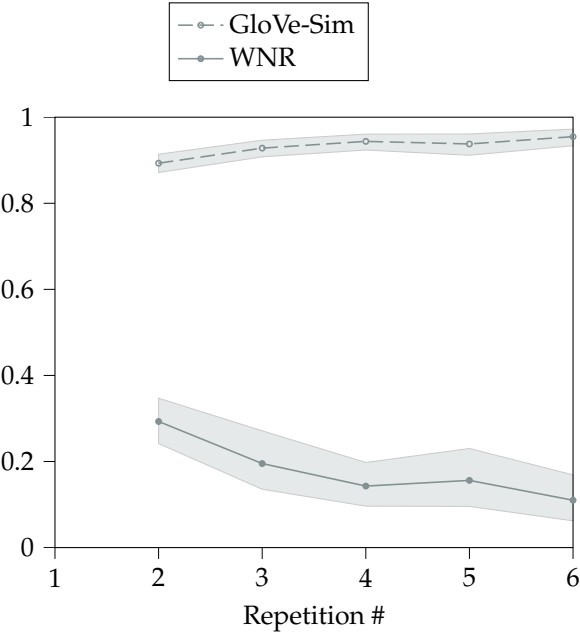

Figure 4: GloVe embedding similarity and WNR between messages from consecutive repetitions. Every increase in GloVe embedding similarity is captured by a corresponding decrease in WNR, and vice versa. Margins of Error are 95% bootstrapped CIs.

We also report a variant of WNR, which is not normalized by length. We refer to this variant as Word Novelty Distance (WND). WND is sometimes more interpretable because each

unit difference in it directly corresponds to a word insertion or deletion. However, because WND is small between any two short messages, it can be harder to interpret when messages are short. Figure 5 reports WND for our speaker experiments.

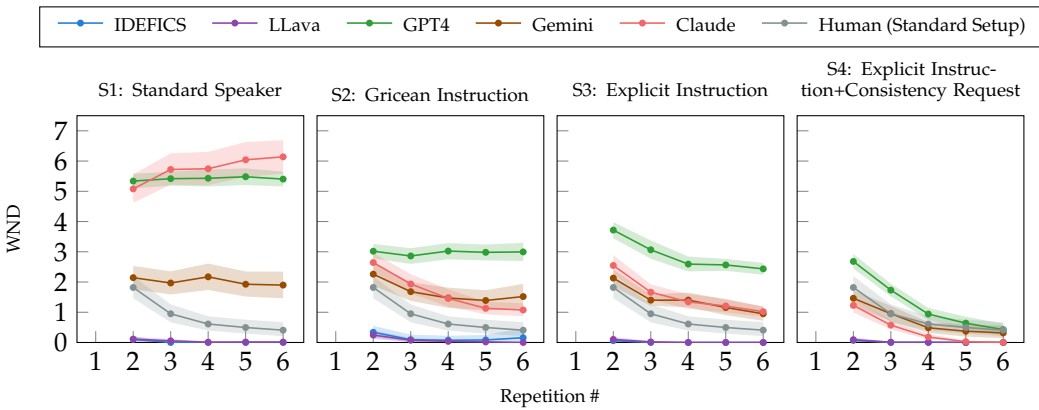

Figure 5: Word Novelty Distance for speaker experiments. Margins of Error are 95% bootstrapped CIs.

## A.3 MLLM Implementation Details

The exact versions of the MLLMs used are idefics-80b-instruct, llava-1.5-13b, gpt-4-1106-vision-preview, Gemini 1.0 Pro Vision, and claude-3-opus-20240229. For IDEFICS, we use 8-bit quantization to fit the 80b model into 3 A6000 GPUs. For all MLLMs, we use a decoding temperature of 0 to avoid the uncertainty caused by sampling, which was also the default for IDEFICS and LLaVa. We use the models' default values for other hyperparameters.

The GPT4 listener used to evaluate model speakers follows the listener interaction setup in L3: Images Once. L3 is the scenario where GPT4 shows its best performance and almost perfectly matches human performance.

LLaVa only supports taking 1 image as input. To bypass this constraint, we merge the 4 images in the referential context into 1 image, using a 2-by-2 grid. Instead of using image labels (A, B, C, D), experiments with LLaVa refer to an original image using its location in the merged image (*top right, top left, bottom right, and bottom left*).

We conduct prompt engineering for each model individually to find the most suitable phrasing of the instructions. The prompt engineering was done over the pilot study dataset released in Hawkins et al. (2020b)'s official Github repository. The pilot study dataset is distinct from the 54 human interactions in ICCA that we use for evaluation. This pilot study dataset contains human-human interactions on easy referential contexts, where the target images are more easily distinguishable to humans with very short messages. For this reason, the human interactions in this pilot study dataset show less language adaptation, and we only use it for prompt engineering and not evaluation.

## B   Example Prompts

---

**Model-as-speaker Prompt**:
[System] Play a game with a listener. This game consists of multiple trials and 4 images (labeled A, B, C, D). You will act as the speaker in this game. In each trial, one of the images is given as the target. You will generate a message to tell the listener which image is the target without mentioning any image label. The listener will try to choose the target correctly based on your message. You will know which image the listener guesses, so you may adjust your messages based on the listener's accuracy. Your reply should only contain the message and be shorter than 20 words. Do not mention any image's label (A, B, C, D) in your message.

Image A: <img1>
Image B: <img2>
Image C: <img3>
Image D: <img4>

Trial 1, the target is Image B.
[Speaker] Message: Two bananas and apples in a white bowl on polka dot cloth.
[System] The listener correctly answered Image B.

Trial 2, the target is Image A.
[Speaker] Message: Bananas on a floral cloth with an Amazon box.
[System] The listener correctly answered Image B.

Trial 3, the target is Image D.
[Speaker] Message: Bananas and apples in a bowl, striped cloth underneath.
[System] The listener mistakenly answered Image C.

Trial 4, the target is Image C.
[Speaker] Message: Assorted fruit in a painted ceramic bowl.
[System] The listener correctly answered Image C.

Trial 5, the target is Image B.
[Speaker] Message:

---

Figure 6: Example prompt for model-as-speaker standard setup. The prompt is shortened and revised for illustrative purposes.

**Model-as-listener Prompt**:
[System] Play a game with multiple trials involving the same set of images. In each trial, I will refer to one of the images with a message. You will guess which image I'm referring to. If present, the history of the previous trials may help you better understand how I refer to specific images.

Trial 1
Image A: <img4>
Image B: <img2>
Image C: <img3>
Image D: <img1>
Which image is this message referring to: The photo with 'amazon' on the back, with 2 bananas.
[Listener] Image D
[System] Correct. I was referring to Image D.

Trial 2
Image A: <img1>
Image B: <img3>
Image C: <img2>
Image D: <img4>
Which image is this message referring to: Photo with a bowl of 3 bananas with a pokadot background.
[Listener] Image B
[System] Wrong. I was referring to Image C.

Trial 3
Image A: <img3>
Image B: <img1>
Image C: <img2>
Image D: <img4>
Which image is this message referring to: A bowl full of mixed fruit, black background.
[Listener] Image A
[System] Correct. I was referring to Image A.

Trial 4
Image A: <img3>
Image B: <img4>
Image C: <img2>
Image D: <img1>
Which image is this message referring to: Photo with a bowl of 3 bananas with stripes and other patterns in the background.
[Listener] Image B
[System] Correct. I was referring to Image B.

Trial 5
Image A: <img3>
Image B: <img2>
Image C: <img4>
Image D: <img1>
Which image is this message referring to: Amazon in back with 2 bananas.
[Listener] Image

Figure 7: Example prompt for model-as-listener standard setup. The prompt is shortened and revised for illustrative purposes.

## C  Figures for Error Analysis

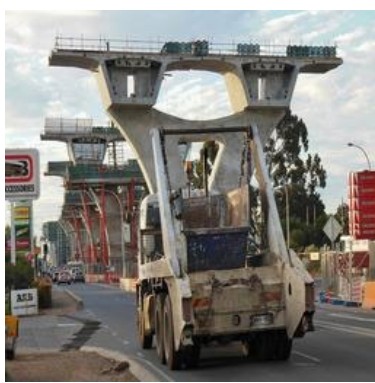

**Human Messages**
1. dirty truck going towards the bridge
2. dirty truck going to bridge
3. how is this even guessing, it's so easy, dirty truck going to bridge
4. dirty truck
5. dirty truck
6. dirty truck

**GPT4 Messages**
1. construction bridge with a concrete mixer truck underneath
2. bridge construction with a white truck
3. bridge work, dirty truck below
4. bridge construction
5. overpass work
6. concrete mixer

Figure 8: Evolution of an image's corresponding messages across 6 repetitions in S3. Both the human speaker and the GPT4 speaker show length reduction, but GPT4's messages are more inconsistent.

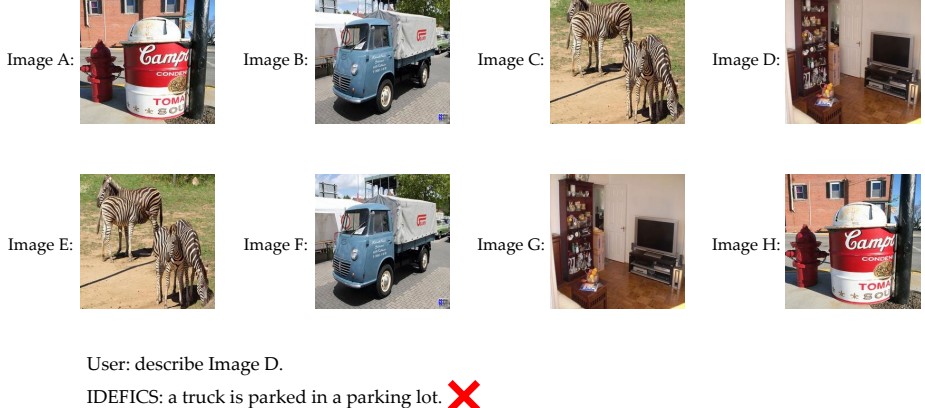

User: describe Image D.

IDEFICS: a truck is parked in a parking lot. ❌

Figure 9: Example of IDEFICS' issue with associating images with their correct labels. The input is a sequence of images interleaved with their labels and a prompt to ask the model to describe *Image [X]*. We tried different formats to label the images, such as * is Image A* and *Image A: * but none helped IDEFICS produce reliable results as the number of images increases.

# D  Image Grounding Impact

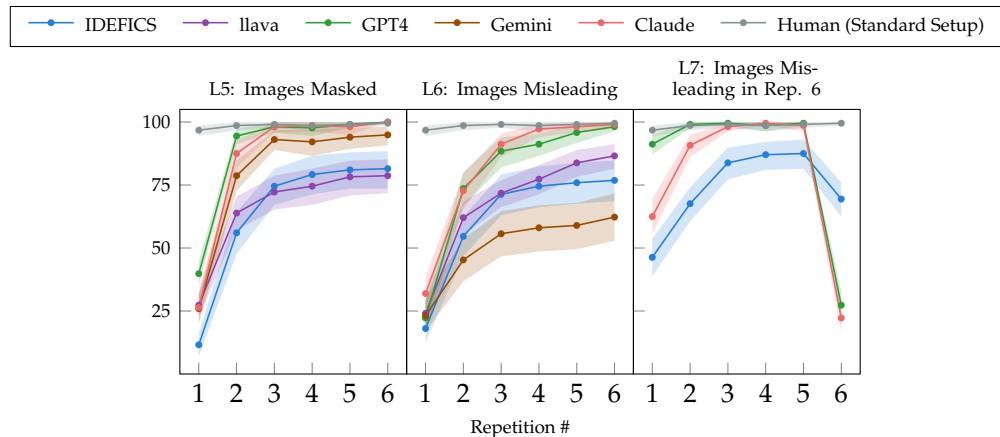

Figure 10: Average accuracy for variants L5-7. Margins of Error are 95% bootstrapped CIs.

The results with L3 (Section 5.1) raised concerns about whether the models are using images effectively or just exploiting label-message association. We develop three variants to study this:

**L5: Images Masked**  Each image is replaced by an image mask, where all pixels have the black color, (0, 0, 0). The image order is persistent, so the same label refers always to the same underlying image that is masked. For the first four trials in Repetition 1, the model has to make a guess. Regardless of the guess, the correct target is given by the system feedback as usual. Theoretically, the model can use the feedback for later repetitions. For a model to succeed in this variant, it must associate the messages with the image labels since no actual images are given. This variant tests the extent to which a model can exploit label-message associations through in-context learning as a way of convention understanding.

**L6: Images Misleading**  This variant complements L5 to further study the impact of text signals and images. The setup here follows L3, showing the referential context once, except that we manipulate the images to be misleading. For the manipulation, we shuffle the images when presenting them to the model listener at the beginning of the game, but we do not change the gold image labels in the system. Therefore, *Image [X]* from the speaker and the system's perspective is likely different from the *Image [X]* from the listener's perspective, and so on. For example, using the context in Figure 11, the model may see the message *Photo with a bowl of 3 bananas with pokadot* and choose Image C but the system and the speaker would always think the image that features bananas and polka dot is Image B, since we shuffled the images without updating the gold labels accordingly. Then the system feedback would be *wrong, the correct answer is Image B*. To succeed under this setting, the model must learn to ignore the images and just associate all the descriptions related to polka dot (for example) with the label *Image B*.

**L7: Images Misleading in Rep. 6**  This variant further tests if the models' previous promising performance in L3 comes from just exploiting the textual signals (the label-message associations) and ignoring the visual input. We show the context once similar to L3 at the beginning, but then show shuffled versions during each trial in the last repetition, without updating the gold labels (same manipulation as L6). We hypothesize that if the model tends to ignore the image input and rely on textual associations in the conversation history, this manipulation will have little impact on its prediction or the accuracy calculated based on the old gold label. This variant requires 20 images in the last trial so it cannot be used with LLaVa or Gemini.

When models cannot utilize the image input (L5 and L6), all models start with around or below random chance accuracies (25%). As the interaction progresses, models exploit past messages and feedback via label-message associations to improve performance and display a trend of improvement. L5 and L6 results together demonstrate that exploiting label-message associations while ignoring the images can easily emerge as an effective mechanism for MLLMs' convention understanding behavior, provided that the image referent has a consistent textual label.

Injecting misleading images in the last repetition (L7) leads to a significant drop in performance relative to previous repetitions for IDEFICS (87.5→69.4%), GPT4 (99.5→27.3%), and Claude (98.6→22.2%). The drop is particularly conspicuous for GPT4 and Claude, where performance goes down to around random chance. This suggests that GPT4 and Claude do not overly exploit the consistent textual associations from the history, because otherwise they would not have been 'misled' by the manipulations in the last repetition. This is desirable because it shows that the visual input matters to these models' output. On the other hand, while IDEFICS also shows a drop in performance, it is not as strong as what the other two models show. This indicates that although IDEFICS is not completely ignoring the visual input, it does exploit the textual associations beyond what is desired.

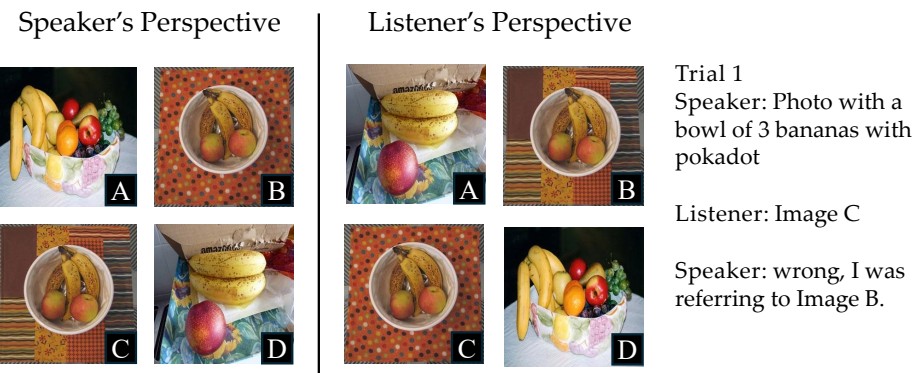

Figure 11: Example Trial 1 from L6: Images Misleading

