# OpenReview forum: "Talk Less, Interact Better: Evaluating In-context Conversational Adaptation in Multimodal LLMs"
_colmweb.org/COLM/2024/Conference — COLM_

### Official Review · Reviewer_8pZ2 · 2024-05-10

**Rating:** 8
**Confidence:** 4
**Ethics Flag:** 1

**Summary:**

This paper presents a series of experiments designed to probe the ability of LLMs to form conceptual pacts / to act on precedent in repeated reference games. Given that LLMs have been tested on reference games before, this is a natural next step to test, and it is good to see it realised. The authors find that there is very little evidence of LLMs treating the conversational context in the same way as the presence of conversational (and hence shared) context would be by natural interlocutors. Only when coaxed through very suggestive prompting, some small effect can be found.
The paper is well written and easy to follow, and could be useful if only to introduce the relevant literature to a field where talk about "mental models" and "theory of mind" is often very cheap.

**Questions To Authors:**

- Can you say more about why you engineered promts for each model? This is a question that arises in all kinds of between-model evaluations, and people come to very different conclusions. But it seems to me that most often, people decide on "no prompt engineering is fairest".

- Not a question, but a comment: You could mention the PhotoBook dataset in the related work, which was built to study exactly your main phenomenon. (Haber et al 2019)

- Not a question, but a comment, part II: You could mention as related the model-model self-play frameworks that came out last year, which pioneered the setup that you are also using. One of which, (Chalamalasetti et al 2023) even includes a reference game as one of the test dialogue games. (But they are not focussing on repeated episodes.)


Kranti Chalamalasetti, Jana Götze, Sherzod Hakimov, Brielen Madureira, Philipp Sadler, and David Schlangen. 2023. clembench: Using Game Play to Evaluate Chat-Optimized Language Models as Conversational Agents. In Proceedings of the 2023 Conference on Empirical Methods in Natural Language Processing, pages 11174–11219, Singapore. Association for Computational Linguistics.

Janosch Haber, Tim Baumgärtner, Ece Takmaz, Lieke Gelderloos, Elia Bruni, and Raquel Fernández. 2019. The PhotoBook Dataset: Building Common Ground through Visually-Grounded Dialogue. In Proceedings of the 57th Annual Meeting of the Association for Computational Linguistics, pages 1895–1910, Florence, Italy. Association for Computational Linguistics.

**Reasons To Accept:**

- Very timely experiment, well conducted
- Well and very clearly written
- Good command of the relevant literature

**Reasons To Reject:**

- Discussion somewhat disappointing. I think the results are stronger than the authors let on: The fact that the models fail so completely to exhibit precedent effects indicates that they, despite what many people seem to hope, have not transcended beyond being text completion models and towards being agent models. The models simply have no reason to communicate efficiently -- among other things, because they are not communicating.

- Related to that, I found the experiments on using prompts to induce the desired behaviour a bit unnecessary. With these prompts, the focus moves from investigating what kind of entity has been learned to less interesting engineering questions. (Although I do understand the rationale that users of these models might even be confused if apparent conceptual pacts are broken.)

---

> ### Author Rebuttal · Authors · 2024-05-30
>
> Thank you for your comments!
>
> >I think the results are stronger than the authors let on...
>
> We will improve and expand the discussion (if accepted, with the extra camera-ready page). The framing you suggest (agent vs next-word predictor) nails it: LLMs have no reason to communicate efficiently (no cost/effort). We hypothesized that they may still acquire it by training on human data. Our results clearly show this is not the case, which can be considered surprising given how much human data is used and how well the models imitate it.
>
> >Experiments on using prompts to induce… bit unnecessary
>
> While we hope adaptation to arise naturally, we need to answer questions about inducing it (e.g., as raised by R-XRWU), and prompting is often considered promising. We also wanted to see if MLLMs do have the ability for Gricean behavior, and maybe just not show it without being “prompted”. Our results show this is not the case, and prompting is not sufficient for such pragmatic behaviors.
>
> More profoundly, these variants reveal underexplored causes of failures. For example, the gain in lexical efficiency without message convergence under some prompts is a surprising finding, which will caution future researchers not to rely on length reduction as the sole indication of efficiency. Overall, to offer a framework for evaluating MLLMs, we wanted ICCA to be thorough and address the different ways people may use MLLMs (e.g., different prompt intervention/interaction formats).
>
> **Engineering prompts for each model**
>
> We wanted to give models the best chance, even at the cost of extra development. For example, a prompt that worked well for one model might lead another model to sometimes output invalid outputs (e.g. “I’m not sure about the answer”), whereas additional instructions against these trivial but adverse behaviors may hurt the models not displaying these behaviors (verbosity of the instruction can hurt LLM performance [1]). By adopting model-specific prompts, we can mitigate these issues. With this being said, all model-specific prompts under the same experiment setup are based on the same instruction. Only the prompts across different interaction variants (different experiment setups) have a substantial difference.
>
> [1] https://arxiv.org/pdf/2307.03172
>
> **Related work**
>
> Thank you for suggesting the related works! We will discuss the photobook dataset, Chalamalasetti et al 2023, and the broad picture of using self-play to evaluate LLMs as conversational agents.

---

> > ### Comment · Reviewer_8pZ2 · 2024-06-05
> > **read**
> >
> > Thank you for your reply. I am still convinced that this paper would make a valuable addition to the conference and the wider literature, and will keep my score.

---

### Official Review · Reviewer_eDRe · 2024-05-11

**Rating:** 9
**Confidence:** 4
**Ethics Flag:** 1

**Summary:**

The paper proposes a framework to evaluate the conversational adaptation of MLLMs. The paper then describes evaluations of a number of MLLMs. The results show that the generations do not become more efficient over time, as would be desirable. However, more careful prompting somewhat mitigates this. These findings have implications for the development of MLLMs.

**Reasons To Accept:**

This is an important problem, and the paper describes a valuable contribution to addressing it. The setup is simple and clear, which I see as a strength. I also found the introduction well-motivated and the background sections very detailed.

**Reasons To Reject:**

Some limitations and issues I wanted to raise include:

-The contribution appears somewhat iterative compared to existing setups and measurements by Hawkins et al (2020ab), extending to evaluating MLLMs. The novelty is not very clearly communicated to readers.

-The paper is very relevant to, but does not engage with the concepts and findings on conversational grounding and building common ground through dialogue. Please see related work [1-4] on some relevant seeds for background and existing studies regarding evaluating and improving conversational adaptation of LLMs (especially 3 and 4, which characterize similar limitations in similar settings).

-Anthropomorphizing language should be avoided, especially when the claims are unfounded ("they (MLLMs) may understand the increasingly efficient language of their interlocutor"; this study doesn't aim to measure understanding); "MLLMs may passively understand" (see [5,6])

1. Clark, Herbert H., and Susan E. Brennan. "Grounding in communication." (1991).
2. Andukuri, Chinmaya, et al. "Star-gate: Teaching language models to ask clarifying questions." arXiv preprint arXiv:2403.19154 (2024).
3. Shaikh, Omar, et al. "Grounding Gaps in Language Model Generations" NAACL 2024.
4. Testoni, Alberto, and Raquel Fernández. "Asking the Right Question at the Right Time: Human and Model Uncertainty Guidance to Ask Clarification Questions." EACL 2024.
5. Inie, Nanna, et al. "From" AI" to Probabilistic Automation: How Does Anthropomorphization of Technical Systems Descriptions Influence Trust?." arXiv preprint arXiv:2404.16047 (2024).
6. Abercrombie, Gavin, et al. "Mirages: On anthropomorphism in dialogue systems." arXiv preprint arXiv:2305.09800 (2023).

---

> ### Author Rebuttal · Authors · 2024-05-30
>
> Thank you for your comments!
>
> **Comparison with Hawkins et al (2020ab)**
>
> We are inspired by Hawkins’ works, rely on their data, and discuss them extensively. We will enhance the discussion of how our work differs. First, we take their data and the reference game scenario in a very different direction, both our object of study and research questions are different. Hawkins presents an approach of adapting through tuning model parameters during an interaction. We focus on diagnosing the behavior of LLMs (a class of models very different from the models Hawkins studied) and on in-context learning/adaptation. Additionally, our evaluation differs. Hawkins focuses only on the standard setup. When a model fails under the standard setup, there can be many contributing factors to the failure. We design variants specifically to diagnose the types and causes of model failures for in-context LLM adaptation. As you suggest, we will communicate these differences more clearly and explicitly.
>
> **Adding related works**
>
> These are great suggestions! We will improve the paper to discuss [1-4] (if the paper is accepted, we get an extra page for the camera ready).
>
> It’s particularly relevant that [3] shows the unintended consequences of using explicit prompts to help form common grounds, which is consistent with our findings and intuition that explicit prompt intervention cannot solve adaptation failures. Discussing [3] will provide a more comprehensive picture of this issue. The synthesis of our work with [3] and [4] also highlights interesting connections. Both [3] and [4] study scenarios where common ground requires explicit acts (e.g., clarification/followup questions) and LLMs presume such common grounds too easily without adequately using those acts. In our case, ad-hoc conventions form without explicit actions in humans, whereas LLMs fail to do so. These different types of common grounds bring about different challenges. This highlights both the richness and the importance of the space.
>
>
> >Anthropomorphizing language should be avoided
>
> We completely agree with the importance of avoiding such language! We will revise the writing.

---

### Official Review · Reviewer_MJyu · 2024-05-11

**Rating:** 4
**Confidence:** 5
**Ethics Flag:** 1

**Summary:**

The manuscript explores the ability of MLLMs to engage in reference games, which involve a duo of a speaker and a listener examining similar images. The speaker's task is to choose and describe one image for the listener to pinpoint. While simple for humans, the game's purpose is to uncover tactics to reduce the length of descriptions needed over time through implicit agreements among the participants. The core experiment involves MLLMs playing these games to see if they can evolve such communicative strategies. The MLLM prompts are specifically designed to encourage this behavior. The study notes that the state-of-the-art MLLM tested did not effectively develop these strategies for more efficient communication, although it could comprehend them when presented.

**Reasons To Accept:**

Overall the paper present a decent piece of work. The presentation is clear and the concepts well introduced and recalled. The study can be a first step on the line to train MLLM with better capacity of adapting or forming ad-hoc conventions when communicating with humans.

**Reasons To Reject:**

The title is misleading: the term adaption is way too vague to describe the phenomenon taking place during reference games. And even more it can let think that the LLMs may be adapted to reach such specific objectives. But the study boils down to pure observation of MLLMs' behaviors when used in reference games. Proposing different prompts with customized instructions ("Gricean" etc) is a good start, but one may straightforwardly thing of means to accompany these prompts and re-enforce their effects (for instance depending on the type of exemples proposed in the context, or using MLLMs in several epochs; including explicit summarisation/simplification taks). Here's what we might like to read.

---

> ### Author Rebuttal · Authors · 2024-05-30
>
> Thank you for your comments!
>
>
> >the term adaption is way too vague to describe the phenomenon taking place during reference games.
>
> We use “adaptation” following established conventions in cognitive science that describe the phenomena humans show in reference games: spontaneously and gradually adopting more efficient language and understanding the interlocutor better. These are exactly the behaviors we are interested in. Our abstract and introduction clarify that our goal is to create tools to evaluate whether MLLMs display such adaptations as humans do, which are essential for communication efficiency. We will make sure the intro relays the term "adaptation" in a concrete and clear way.
>
> Also, such adaptations are broad phenomena, and not just about reference games. But, they are known to be well-demonstrated, measurable, and extensively studied in cognitive science through reference games. If LLMs acquire linguistic abilities similar (or better) to humans, reference games make for an ideal testbed.
>
>
> >And even more it can let think that the LLMs may be adapted to reach such specific objectives.
>
> The title specifically focuses on “in-context” behaviors of MLLMs, so it clearly states the kind of adaptation we are studying, meaning: in contrast to tuning parameters or changing the architectures. In-context learning is well established in the study of LLM as a way to modify (i.e., adapt) LLM behavior. We also discuss the relatedness of in-context adaptation and few-shot in-context learning in the related work section. We will further clarify this distinction as early in the paper as possible.
>
>
> >one may straightforwardly think of means to accompany these prompts and re-enforce their effects...
>
> Our focus is evaluating if LLMs without further modifications display this behavior, to see if it arises from their training. This is key to displaying this behavior in a truly general way. Our goal here is not just to understand the abilities of LLMs, but to contribute to the fundamental understanding of what arises (and what does not) from the way LLMs are trained. This is why we made prompting efforts specifically to elicit behavior that is already baked into the model, but maybe doesn’t show for one reason or another. Building on top of LLMs to further address their deficiency in adaptation is a good direction for future work, one that is enabled and supported by the ICCA framework.

---

### Official Review · Reviewer_XRWU · 2024-05-12

**Rating:** 6
**Confidence:** 4
**Ethics Flag:** 1

**Summary:**

In order to understand the Multi-modal LLM (MLLM)'s capability of In-context Conversational Adaption (ICCA). This paper defined ICCA as a framework to evaluate.

ICCA includes
(1) A human-human interaction dataset that focus on repeated reference game interaction
(2) A few success metrics including accuracy, message length, and Word Novelty Distance (WND)
(3) Two types of set up: model-as-speaker and model-as-listener with different types of prompt settings

With such a framework, a series of experiments has been conducted and the results suggested the SOTA MLLMs are not good at in-context adaptation yet.

The paper is structured well, and quite easy to follow. The experiment design looks good to me. The result itself isn't surprising. Thus I am on the borderline leaning towards accepting the paper.

**Reasons To Accept:**

**Novelty**
Building metrics to evaluate LLMs on in-context adaptation is a nice thing to have. When LLMs are more capable of getting answers right, the next step might be getting the communication more efficient during multimodal settings.

**Clarity**
The paper is clear about what it does and how things are being done.

**Sounds Approach**
The approach of the paper is sound. Defining

**Reasons To Reject:**

**Significance**
The finding of the paper isn't surprising. Establishing the metrics and building a baseline about MLLMs on one task itself is a contribution, I don't feel excited about the result and the work.

---

> ### Author Rebuttal · Authors · 2024-05-30
>
> Thank you! We are glad you appreciated our novelty, clarity, and soundness.
>
> **Significance of the results**
>
> Are our results surprising? The progress of LLMs has been impressive, with new results often surpassing human performance. LLMs are trained on human language data, which is rich in adaptation and convention formation/use. Given these models’ success, one would expect them to acquire these characteristics of human language as well. Our results show the opposite: while some aspects of language are well-addressed by current paradigms, others are not. Beyond the fundamental learning problem, our work’s importance was made clear by the recent conversational use cases LLM companies have been demoing (e.g., OpenAI’s latest demo). Overall, this illustrates that the problem is both important and underdiagnosed by current research. ICCA is a framework to drive important future development, in a way that was not previously on the agenda or easy to diagnose. We will improve the presentation of the significance in the paper.
>
> **Breadth & depth of the results**
>
> Indeed, we observe MLLMs do not display in-context conversational adaptation. Our analysis and ICCA itself go deeper though, including:
> * Even with explicit instruction on length reduction (Variant S3), the models display lexical efficiency but miss critical aspects of true communication efficiency (convergence and consistency). This finding is unexpected (contrary to human behaviors). It emphasizes that communication efficiency is not only about length reduction. ICCA gives future researchers the tools to look beyond length.
> * Some models can display seemingly good listener accuracy by memorizing textual information without forming conventions on visual signals. Our different interaction variants make ICCA an effective tool for such diagnosis. This is not possible using the standard evaluation from the prior work. We discuss this in Appendix D, but will expand the discussion in the paper.
> * Associative recall, the ability to recall information mentioned in-context is believed to be predictive of in-context learning quality [1]. We observe underexplored implications of this ability. Throughout an interaction, IDEFICS and LLaVa assign high probability to the exact repetition of initially used phrases, marking a good associative recall. But this hinders the development of more concise phrases. We will further elaborate on this issue when we revise the paper.
>
> [1] https://arxiv.org/pdf/2312.04927

---

### Decision · Program_Chairs · 2024-07-10

**Decision:**

Accept

**Comment:**

This paper proposes a framework called ICCA (In-context Conversational Adaptation) to evaluate the ability of Multi-modal Large Language Models (MLLMs) to engage in reference games and develop efficient communication strategies over time. The authors include experiments on the human-human repeated reference game dataset from Hawkins et al (2020), and compare different setup variants. Experiments using the ICCA framework suggest that current state-of-the-art MLLMs do not effectively adapt their communication strategies in-context: they are relatively good "listeners" but not human-like "speakers".

The majority of reviewers agree that this paper addresses an important problem and makes a valuable contribution to understanding the in-context adaptation capabilities of MLLMs. The ICCA framework provides a clear and well-motivated approach to evaluate these models' ability to develop efficient communication strategies over time, drawing inspiration from human behaviors in reference games. I personally loved this paper.

Key strengths highlighted by the reviewers include the novelty of the evaluation framework, the clarity of the paper's presentation, and the soundness of the experimental approach. The findings, while perhaps not surprising, are significant in showing that current MLLMs do not effectively display the kind of in-context adaptation that humans exhibit in these scenarios. This has important implications for the development and deployment of these models in conversational settings.

The authors provide thoughtful responses to the reviewers' comments and suggestions in their rebuttal. They clarify the differences between their work and prior related research and propose to enhance the discussion of these differences in the paper. The rebuttal also addresses concerns about the use of model-specific prompts and the inclusion of prompt variants in the experiments, providing convincing justifications for these choices.

In summary, the ICCA framework and the insights gained from its application to MLLMs make this paper accept. The work advances our understanding of the strengths and limitations of these models in conversational settings and lays the groundwork for future research in this area. With some revisions to clarify the novelty and significance of the contributions, as well as to incorporate additional related work and refine the discussion, this paper is likely to have a positive impact on the field.